# Small Brown Planthopper Nymph Infestation Regulates Plant Defenses by Affecting Secondary Metabolite Biosynthesis in Rice

**DOI:** 10.3390/ijms24054764

**Published:** 2023-03-01

**Authors:** Shuai Li, Liangxuan Qi, Xinyang Tan, Shifang Li, Jichao Fang, Rui Ji

**Affiliations:** 1Jiangsu Key Laboratory for Food and Safety-State Key Laboratory Cultivation Base of Ministry of Science and Technology, Institute of Plant Protection, Jiangsu Academy of Agricultural Sciences, Nanjing 210014, China; 2State Key Laboratory for Biology of Plant Diseases and Insect Pests, Institute of Plant Protection, Chinese Academy of Agricultural Sciences, Beijing 100193, China; 3Key Laboratory for Conservation and Use of Important Biological Resources of Anhui Province, Anhui Provincial Key Laboratory of Molecular Enzymology and Mechanism of Major Diseases, College of Life Sciences, Anhui Normal University, Wuhu 241000, China

**Keywords:** small brown planthopper, rice, plant-insect interaction, transcriptome, metabolome

## Abstract

The small brown planthopper (SBPH, *Laodelphax striatellus*) is one of the most destructive insect pests in rice (*Oryza sativa*), which is the world’s major grain crop. The dynamic changes in the rice transcriptome and metabolome in response to planthopper female adult feeding and oviposition have been reported. However, the effects of nymph feeding remain unclear. In this study, we found that pre-infestation with SBPH nymphs increased the susceptibility of rice plants to SBPH infestation. We used a combination of broadly targeted metabolomic and transcriptomic studies to investigate the rice metabolites altered by SBPH feeding. We observed that SBPH feeding induced significant changes in 92 metabolites, including 56 defense-related secondary metabolites (34 flavonoids, 17 alkaloids, and 5 phenolic acids). Notably, there were more downregulated metabolites than upregulated metabolites. Additionally, nymph feeding significantly increased the accumulation of seven phenolamines and three phenolic acids but decreased the levels of most flavonoids. In SBPH-infested groups, 29 differentially accumulated flavonoids were downregulated, and this effect was more pronounced with infestation time. The findings of this study indicate that SBPH nymph feeding suppresses flavonoid biosynthesis in rice, resulting in increased susceptibility to SBPH infestation.

## 1. Introduction

Rice (*Oryza sativa*) is one of the most important grains in China, and its production is crucial to food security. The rice production is threatened by various pests [1,2], such as the small brown planthopper (SBPH, *Laodelphax striatellus*), brown planthopper (BPH, *Nilaparvata lugens*), and white-backed planthopper (WBPH, *Sogatella furcifera*), which are typical and destructive piercing-sucking insects that feed from the phloem, damage rice, and transmit multiple viral diseases [3,4].

The application of chemical pesticides is the primary strategy to control rice planthoppers; however, their long-term and heavy use pose dangerous risks to beneficial natural enemies and pollinators, enhancing the resistance of insects to pesticides, leading to environmental contamination, and affecting ecological equilibrium. Achieving sustainable management of planthoppers requires an understanding of the molecular basis of pest outbreaks, particularly the mechanisms of plant-insect interactions [5]. The identification and utilization of resistance factors in rice are safe, economical, and effective ways to control planthoppers. Furthermore, understanding the systematic transcriptomic and metabolic responses of rice to planthopper infestation can reveal the defense mechanism against planthoppers and help identify resistance genes or metabolites that are beneficial for rice resistance breeding and the development of environmentally friendly pest control strategies [6].

For protection against herbivores, plants have evolved complex constitutive and inducible defenses. These defenses can be categorized as physical structures (e.g., spines, thorns, and trichomes) and/or chemical components (e.g., all classes of secondary metabolites) that are either constitutively produced by the plant or specifically induced after a herbivore attack [7]. Plant secondary metabolites are classified into three types based on their chemical composition, structure, and biosynthetic pathways: phenolic compounds (phenolic acids and flavonoids), nitrogen/sulfur-containing compounds (alkaloids and glucosinolates), and terpenes/terpenoids [8], all of which play crucial roles in plant defense against biotic and abiotic stresses [9]. Transcriptomic analysis is a powerful strategy for investigating the expression patterns of metabolite biosynthetic genes, whereas metabolomic analysis directly detects the metabolite content, reflecting the metabolic state of an organism or cell [10]. The combination of transcriptomic and metabolomic analyses can help reveal the molecular biological processes of plant chemical defense [11]. Recent studies have profiled the transcriptome and metabolome of rice in response to herbivores. *Chilo suppressalis* infestation induces the release of rice volatiles known to be attractive to planthoppers, and a metabolomic analysis revealed that increased free amino acid content and reduced sterol content in rice damaged by *C. suppressalis* promote planthopper growth [12]. Additionally, rice volatiles induced by *C. suppressalis* infestation have a significant repellent effect on the egg parasitoid *Anagrus nilaparvatae*, the natural enemy of BPH [13].

Dynamic changes in the transcriptome and metabolome of rice in response to planthopper adults have been reported. BPH adult infestation causes drastic metabolic changes that promote phenylpropane biosynthesis, flavonoid biosynthesis, and plant hormone signal transduction [14]. In the laboratory and in the field, BPH female adults infestation decreased the expression of rice DELLA protein OsSLR1, increased the constitutive levels of defense-related compounds, phenolic acids, lignin and cellulose, as well as the resistance of rice to BPH [15]. Additionally, the major impacts of female planthopper adults on plants include oviposition and feeding [16]. The protein profiles of SBPH-resistant (Pf9279-4) and SBPH-susceptible (02428) rice genotypes were compared in response to SBPH female infestation. The differentially expressed proteins were primarily involved in protein metabolic processes, energy metabolism, amino acid metabolism, and transcriptional regulation. Furthermore, isoflavone reductase-like protein, the key enzyme in isoflavone synthesis, was upregulated in Pf9279-4 compared to 02428 [4,17]. However, the impact of planthopper nymph feeding on the rice metabolome remains unknown.

This study aimed to identify genes and metabolites associated with the adaptation of rice to SBPH nymph feeding. We compared SBPH performance in nymph-pre-infested and uninfested rice plants, subsequently analyzed the transcriptome and metabolome profiling of rice infested by SBPH nymphs, and investigated the functions of defense-related genes and secondary metabolites in rice.

## 2. Results

### 2.1. The Susceptibility of Rice to SBPH Infestation Induced by SBPH Nymph Pre-Infestation

We performed choice and no-choice tests to determine whether SBPH nymph pre-infestation can regulate plant defenses and influence the susceptibility of rice plants to subsequent SBPH infestation. In the choice test, both nymphs and adult female SBPHs preferred the pre-infested rice plants to the control plants (Figure 1A,B). In the no-choice test, we released female adults on plants pre-infested with nymphs and control plants for two days and then observed their fecundity. The number of eggs laid by SBPHs on the pre-infested rice was significantly higher than that on the control rice plants (Figure 1C). These findings suggest that SBPH pre-infestation increases the susceptibility of rice to SBPH.

### 2.2. Responses of Rice to SBPH Nymph Infestation at the Transcriptional Level

We aimed to identify valuable defense-related genes and metabolites in rice adaptation to SBPH nymph feeding by analyzing the underlying mechanisms of the interactions between SBPH nymphs and rice at the transcriptional level. The transcriptomes of SBPH-infested and uninfested plants were characterized using Illumina RNA sequencing at various infestation times (0, 12, 24, and 48 h). Three SBPH-infested rice groups (trans12, trans24, and trans48) and control samples (trans0) were collected for transcriptome analysis. The high proportion of reads with Qphred (Q) 30 scores (≥92.75%) for each library indicated the high quality of the RNA sequencing and the reliability of the data (Appendix A). The differences between groups were greater than the differences within groups, according to principal component analysis; gene expression pattern correlation analysis between biological replicate samples (≥0.96) indicated a high degree of reproducibility (Figure 2A and Appendix A).

The centralized and normalized FPKM expression of the differentially expressed genes (DEGs) was extracted to construct a hierarchical cluster analysis (HCA) map (Figure 2B). We observed some significant differences between the SBPH-infested rice groups and the control group. We identified a total of 2053 (528 upregulated and 1525 downregulated) in the 12 h time-point compared with the controls at time 0 (12-trans), 3398 (1155 upregulated and 2243 downregulated) in the 24 h time-point compared with the 0 h (24-trans), and 3056 DEGs (1186 upregulated and 1870 downregulated) in the 48 h time-point compared with the 0 h (48-trans) (Figure 2C and Appendix A). As a result, the differences were the greatest at trans24, indicating a significant transcriptomic response to SBPH infestation under these conditions. Additionally, we compared the DEGs of the 12-, 24-, and 48-trans comparison crowds and identified 1499 DEGs that were co-expressed in all three groups (Figure 2D).

Kyoto Encyclopedia of Genes and Genomes (KEGG) pathway enrichment analysis of all the DEGs revealed 23 pathways, including plant–pathogen interaction, ABC transporters, and the biosynthesis of plant secondary metabolites, such as diterpenoid flavonoids, flavones, flavonols, and phenylpropanoids, to be significantly enriched at all test time-points. Additionally, the cutin, suberin, and wax biosynthesis pathways, ubiquinone and other terpenoid-quinone biosynthesis, and starch and sucrose metabolism were slightly enriched in the 48-trans group. The top 20 enriched pathways in the comparison of the three groups are shown in Figure 3A, Appendix A, and Appendix A. GO analysis revealed that the DEGs were mostly involved in biological processes, with significant enrichment in catalytic activity, cellular anatomical entities, cellular processes, metabolic processes, responses to stimuli, and biological regulation (Figure 3B, Appendix A and Appendix A). Overall, these findings demonstrated that SBPH infestation influenced various biochemical processes.

Ultimately, fourteen genes, including those involved in flavonoid biosynthesis (five genes), plant–pathogen interaction (three genes), diterpenoid biosynthesis (three genes), and three other genes (Os09g0403300, Os12g0441600, and Os12g0559200), were randomly chosen from candidate DEGs and examined to validate the RNA-seq data by RT-qPCR. The findings were broadly consistent, indicating that the transcriptome data were reliable (Appendix A).

### 2.3. Metabolic Responses of Rice to SBPH Nymph Infestation

To gain a more comprehensive understanding of the influence of nymph feeding on rice, we performed metabolic analyses using broadly targeted metabolomic technology. We detected 451 metabolites in the leaf sheath samples, of which 92 showed differential accumulation, including 56 plant secondary metabolites (34 flavonoids, 17 alkaloids, and 5 phenolic acids), 18 lipids, 5 nucleic acids, 3 organic acids, 2 amino acids, 1 lignan, and 7 others (Figure 4A and Appendix A). Principal component analysis was used to identify metabolic changes following SBPH nymph infestation in rice. A distinct difference was observed in the PCA score plot between the metabolites from the SBPH-infested groups and the control (meta0). The two principal components (1 and 2) separated the two sample groups and accounted for 39.61% and 17.69% of the total variation in the entire data set, respectively. The dimensional separations on both axes revealed distinct differences in the metabolite content between the comparison groups (Figure 4B). Overall, most differentially accumulated metabolites (DAMs) were secondary metabolites, and the number of downregulated DAMs was higher than that of upregulated DAMs.

Metabolite data from samples obtained at the 24 h time-point compared with the controls at time 0 (meta24 vs. meta0) revealed 10 DAMs with 6 defense-related secondary metabolites upregulated and 47 DAMs with 32 defense-related secondary metabolites downregulated. There were 78 significant DAMs at the 48 h time-point compared to the 0 h (meta48 0 vs. meta0), with 14 DAMs (including 11 defense-related secondary metabolites) upregulated and 64 DAMs (including 38 defense-related secondary metabolites) downregulated. In the 72 h time-point compared with the 0 h (meta72 vs. meta0), 70 DAMs were screened, of which 6 DAMs, including 1 defense-related secondary metabolite, were upregulated and 64 DAMs, including 39 defense-related secondary metabolites, were downregulated (Figure 4C). The Venn diagram showed 47 DAMs co-expressed in all three comparison groups. Comparing the meta48 group to the other groups revealed that a higher number of DAMs were induced, suggesting a more significant metabolic response to SBPH infestation at this time-point (Figure 4D).

To investigate the differences in the identified DAMs in response to SBPH nymph infestation, we selected the meta48 0 vs. meta0 comparison group to conduct differential metabolite co-expression analysis. Among the upregulated metabolites, defense-related alkaloids and phenolic acids (PAs) were significantly more accumulated in SBPH-infested samples than in the controls. The alkaloids accounted for a large proportion of the upregulated metabolites (50%, 7/14) and included N-Z-p-coumaroyl-tyramine (16.97-fold), N-cis-paprazine (15.70-fold), N-cis-sinapoyltyramine (15.29-fold), N-trans-feruloyltyramine (9.24-fold), N-cis-feruloyltyramine (8.50-fold), methoxy-N-caffeoyltyramine (7.76-fold), and N-feruloyltyramine (6.61-fold). In contrast, flavonoids and lipids were significantly less accumulated in SBPH-infested samples and accounted for a large proportion of the downregulated metabolites (68.8%, 44/64), including prunetin (0.04-fold), acacetin (0.04-fold), biochanin A (0.04-fold), genkwanin (0.05-fold), palmitoleic acid (0.26-fold), cis-10-heptadecenoic acid (0.31-fold), LysoPE 18:1 (2n isomer) (0.31-fold), and 11 octadecanoic (vaccenic) acid (0.44-fold) (Figure 4A, Appendix A). Subsequently, we analyzed the trend of the relative content of DAMs in different treatment groups using K-means clustering, which revealed nine major trends for all DAMs (Figure 5A, Appendix A). Most defense-related secondary metabolites in clusters 1, 3, 4, 6, and 7 were flavonoids, which were downregulated in the three stages of rice compared to the control group (meta0), whereas clusters 2 and 8 mainly included defense-related alkaloids and PAs that were upregulated in the three treatment groups. These defense-related secondary metabolites demonstrated similar expression patterns in all treatment groups. Furthermore, we subjected the identified DAMs to KEGG pathway analysis, and the significantly different pathways were mainly enriched in the biosynthesis of secondary metabolites, such as flavonoids, isoflavonoids, benzoxazinoids, and galactose metabolism, phosphatidylinositol signaling system, and alpha-linolenic acid metabolism (Figure 5B and Appendix A). Overall, these findings demonstrate that the secondary metabolite profile of rice was altered after SBPH nymph infection, suggesting its crucial role in the rice defense response.

### 2.4. Correlation Analysis between the Transcriptome and Metabolome

To investigate and comprehend the potential regulatory network between DEGs and DAMs, we performed a correlation analysis. In a nine-quadrant graph, quadrants 1 and 9 indicated that the differential expression patterns of genes and metabolites were opposite; quadrant 5 indicated that neither genes nor metabolites were differentially expressed; quadrants 2, 4, 6, and 8 indicated that metabolite expression remained unchanged while genes were up or downregulated, or that gene expression remained unchanged while metabolites were up or downregulated; quadrants 3 and 7 indicated that the differential expression patterns of genes and metabolites were consistent [18]. The nine-quadrant graph revealed that the majority of DEGs were consistent with the DAMs patterns; the greater proportion of genes and metabolites in quadrant 7 indicated that positive gene regulation was dominant over negative regulation in genes affecting metabolic changes (Figure 6A, Appendix A, and Appendix A).

Following correlation analysis, we discovered that several DEGs generated at the 24 h time-point and DAMs collected at the 48 h time-point were enriched in the same KEGG pathways, including flavonoid biosynthesis, metabolic pathways, glutathione metabolism, and glycolysis/gluconeogenesis (Appendix A). Particularly, flavonoids made up a large portion of the downregulated DAMs; nevertheless, alkaloids and phenolic acids made up a large portion of the upregulated DAMs in SBPH-infested samples. This implies that, following nymph infestation, rice’s flavonoid content decreases, while its alkaloids and phenolic acid content increase (Figure 4A, Appendix A). As a result, we focused on the differences in the expression of genes relevant to these secondary metabolite biosynthetic pathways. Among the identified DAMs, the content of 29 flavonoids showed significant decreases of 52–96% in SBPH-infested rice samples compared to the control group (VIP ≥ 1.0, Figure 4A, Appendix A). However, only about half the amount of flavonoids (16/29) was annotated in the flavonoid biosynthesis-related pathways, including flavonoid biosynthesis (14/29, ko00941): 4,2′,4′,6′-Tetrahydroxychalcone (mws1140), pinobanksin (mws0914), butin (pme3475), naringenin chalcone (pme2960), tricin O-sinapoylhexoside (pmb0738), tricin 7-O-feruloylhexoside (pmb0725), tricin 4′-O-(β-guaiacylglyceryl) ether O-hexoside (pmb0723), taxifolin (mws0044), hesperetin 5-O-glucoside (pme1598), dihydrokaempferol (mws1094), tricin O-feruloylhexoside O-hexoside (pmb0721), tricin 5-O-β-guaiacylglycerol (pmb0717), naringenin (pme0376), and hesperetin (mws0463); and flavone and flavonol biosynthesis (6/29, ko00944): tricin 7-O-feruloylhexoside (pmb0725), quercetin 3-O-glucoside (pme3211), acacetin (mws0051), tricin O-sinapoylhexoside (pmb0738), tricin 5-O-β-guaiacylglycerol (pmb0717), and tricin O-feruloylhexoside O-hexoside (pmb0721) (Appendix A).

As shown in Figure 6B, almost all metabolites participating in the flavonoid biosynthesis pathway were downregulated in SBPH-infested groups compared to the control (meta0), and with the increase in infestation time, the reduction in flavonoid metabolites in rice was gradually more pronounced. Additionally, the transcriptome data from the three comparison groups that were involved in this pathway were relatively consistent; while the number of upregulated and downregulated DEGs was equal, the magnitude of upregulated genes was considerably lower than that of downregulated genes, particularly for genes encoding flavanone 3-hydroxylase (Os02g0767300), chalcone synthase (Os04g0103900), naringenin,2C2-oxoglutarate 3-dioxygenase (Os04g0662600), 0-methyltransferase (Os04g0104900), n-methyltransferase (Os12g0202700), and transferase family protein (Os11g0643100) in rice, which showed differentially repressed expression as a result of SBPH infestation, affecting flavonoid biosynthesis (Figure 6C). The correlation network represented the direct relationship between DEGs and DAMs, and there were 13 DEGs associated with 8 flavonoids in flavonoid biosynthesis (ko00941) and 7 DEGs involved in flavone and flavonol biosynthesis (ko00944) (Appendix A and Appendix A). Most DEGs were positively correlated with the accumulation of flavonoids (i.e., Os11g0530600 and Os12g0202700); however, a few were negatively correlated (i.e., Os10g0320100 and Os10g0196100), which reflects the complexity of the gene regulation of flavonoid synthesis. These findings indicate that SBPH nymph infestation is a critical negative modulator of flavonoid biosynthesis in rice.

Seven phenolic acids (PAs) (N-Z-p-coumaroyl-tyramine, N-cis-paprazine, N-cis-sinapoyltyramine, N-trans-feruloyltyramine, N-cis-feruloyltyramine, methoxy-N-caffeoyltyramine, and N-feruloyltyramine) were found to be considerably upregulated in SBPH-infested samples. Notably, these PAs were among the most highly upregulated metabolites (by 6.61–16.97 folds), indicating that SBPH feeding preferentially elicited the accumulation of PAs in the leaf sheath of rice. Additionally, we used the metabolite data collected at 48 h time-point-transcriptome data generated at 24 h time-point compared with the 0 h (meta48-trans24 vs. meta0-trans0) conjoint analysis group as a representative of the three conjoint analysis groups. Three types of PAs (pyrocatechol, coniferyl alcohol, and sorbic acid) were significantly increased among the upregulated metabolites in SBPH-infested plants compared to the control. Among them, pyrocatechol increased the most (mws1358, 3.72-fold). Furthermore, two of these PAs were related to KEGG pathways (ko01100, ko01110, ko00940, and ko00999), such as metabolic pathways (pyrocatechol) and phenylpropanoid biosynthesis (coniferyl alcohol, nws0093) (Figure 4A, Appendix A). In the correlation network analysis, 215 DEGs were associated with pyrocatechol biosynthesis, and 21 DEGs were associated with coniferyl alcohol biosynthesis (Appendix A and Appendix A), reflecting the complexity of gene regulation of metabolite synthesis. These findings demonstrate the existence of a complex regulatory network between the accumulation of secondary metabolites and the level of gene expression, and further research is needed to investigate and validate these key genes involved in flavonoid biosynthesis and PAs in rice.

## 3. Discussion

Plants produce various secondary metabolites in response to biotic and abiotic stresses. However, the metabolites related to defense responses against insect infestation in rice remain largely unclear. In the present study, we investigated the transcriptome and metabolome of rice infested by SBPH nymphs to increase our knowledge of the chemical defense of rice against herbivores. We observed that the transcriptome at 24 h and the metabolome at 48 h comprised the most significant time-points for DEGs and DAMs in response to SBPH nymph infestation in rice, respectively (Appendix A). Additionally, the content of 29 flavonoids was significantly decreased, whereas that of eight alkaloids and four phenolic acids was increased in SBPH-infested rice plants, implying that these components are secondary metabolites closely related to defense against insect infestation in rice.

### 3.1. PAs May Not Play a Key Role in the Adaptation of Rice to SBPH Nymph Infestation

PAs are the main components of the phenylpropanoid pathway [19], exhibiting anticancer, antinociceptive, antiphlogistic, antiviral, antibacterial, and antioxidant properties [20,21,22,23]. They can act as non-enzymatic antioxidants that alleviate the harmful effects of pesticides and free radicals, and play direct defensive roles in plants [24,25]. PAs preferentially accumulate in the young vegetative and reproductive tissues of *Nicotiana attenuata* and are induced by viral infections [26,27]. They are also induced by UV in rice and exhibit anti-*Pseudomonas syringae* properties in tomatoes. The levels of PAs in soybeans are increased in response to herbivore infestation and are related to the palatability of herbivores [28,29]. Two PAs (p-coumaroylputrescine and feruloylputrescine) accumulate in response to infestation by the lawn armyworm (*Spodoptera mauritia*), rice skipper (*Parnara guttata*) larvae, and BPH in rice [30,31,32]. The upregulation of two PAs (phenylpropanoid enzyme and phenylalanine ammonia-lyase) in SBPH-infected rice is correlated with the resistance of different rice varieties [33]. Additionally, PAs are upstream metabolites in the lignin biosynthesis pathway [34], and their content is negatively correlated with lignin [35]. Consistent with these previous findings, we found that lignins and coumarins (pme3140, 6-Hydroxy-4-methylcoumarin) were prominently downregulated (0.14-fold) in SBPH-infested rice plants and were negatively correlated with the upregulated PAs (Appendix A). Furthermore, the accumulation of lignin can enhance the thickness and hardness of the cell wall, blocking the planthopper stylet from penetrating the cell wall and feeding on the phloem sap, suggesting that the increase in PA content may decrease the level of the lignins, and affect the defense mechanism against SBPH in rice [15,36]. In the present study, the significantly accumulated PAs were only detected in the meta48 group, indicating that they may be induced shortly after SBPH nymph infestation. No PAs were identified in the enriched KEGG pathway (Appendix A), implying that PAs do not play a key role in the adaptation of rice to SBPH nymph infestation.

### 3.2. The Suppression of SBPH Nymph Feeding on the Rice Defense Could Be Caused by the Decreased Accumulation of Flavonoids

Flavonoids are an abundant subgroup of polyphenols in various plants. They are synthesized from the starting compound flavanone and classified into several subgroups based on their chemical structures, including flavones, flavanols, flavanones, chalcones, dihydrochalcones, anthocyanidins, and dihydroflavonols [8,37]. Most flavonoids have strong antioxidant activity and play key roles in plant responses to various stresses [38]. Dendrobium viroid infection significantly decreases the accumulation of flavonoids in infected *Dendrobium officinale* stem tissue, and the decreases in these metabolites affect the medicinal components of plants, weakening the host antiviral immune response [39]. Furthermore, flavonoids can result in avoidance and toxicity to insects. Flavonoids isolated from cotton plants can inhibit the growth and pupation of *Helicoverpa armigera* and *Heliothis virescens*, and those isolated from wheat can significantly inhibit the growth, development, and reproduction of *Sitobion avenae* [40]. Additionally, the increased flavonoid content improves the resistance of rice to BPH; the flavanone 3-hydroxylase (*OsF3H*) gene in the flavonoid biosynthetic pathway plays a positive role in mediating both flavonoid biosynthesis and resistance to BPH [41]. Planthopper infestation causes metabolic changes in flavonoid profiles. DAMs belonging to flavonoids were downregulated in BPH-susceptible rice (Nipponbare) but upregulated in the BPH-resistant variety (C331) [14].

The protein profiles of SBPH-resistant (Pf9279-4) and SBPH-susceptible (02428) rice genotypes were compared in response to SBPH female infestation in previous studies. The differentially expressed proteins were primarily involved in the stress response, protein metabolic processes, energy metabolism, amino acid metabolism, and transcriptional regulation. Isoflavone reductase-like protein, the key enzyme in isoflavone synthesis, was upregulated in Pf9279-4 compared to 02428 [4,17]. Consistent with these reports, in our study, 29 flavonoids were downregulated whereas only 1 flavonoid was upregulated in SBPH-infested rice samples (Appendix A), indicating that the rice defense against SBPH nymphs was likely suppressed by the decreased accumulation of flavonoids. In the follow-up work, we will focus on this up-regulated protocatechuic aldehyde (pme2482) and overexpress or knock out this encoded gene in rice to examine the function of this flavonoid against SBPH infection.

### 3.3. Interactions between the Rice Flavonoids and Planthopper Salivary Effectors

Tricin (5,7,4′-trihydroxy-3′,5′-dimethoxy flavone), one of the most important flavonoids, is commonly found in rice plants and can defend against infestation by planthoppers. Tricin stimulation can significantly alter the expression of some metabolism proteins, which are involved in the salivary secretion system, energy metabolism, and carbohydrate metabolism pathways, in the BPH salivary gland [42]. In contrast, BPH feeding reduces tricin accumulation in rice. Additionally, BPH salivary protein 7 (NlSP7) is highly expressed in response to tricin. Knocking down NlSP7 promotes the tricin content in rice plants and alters the feeding behavior as BPH individuals spend more time in the non-penetration and pathway phases and less time feeding on the phloem of rice plants. As a result, NlSP7 functions as an effector that suppresses tricin accumulation in rice [43]. In the present study, four tricins were significantly downregulated in SBPH-infested rice: tricin 7-O-feruloylhexoside, tricin O-feruloylhexoside O-hexoside, tricin 5-O-β-guaiacylglycerol, and tricin O-sinapoylhexoside (Appendix A). This effect may be attributed to the suppression of the accumulation of tricins in rice by salivary effectors, such as SP7 secreted by SBPH nymphs, resulting in increased rice susceptibility to SBPH infestation.

## 4. Materials and Methods

### 4.1. Plant Growth and Insect Rearing

The rice genotype used in the present study was Nipponbare. Rice seeds were pre-germinated in plastic bottles in a greenhouse under conditions of 28 ± 1 °C and a 16 h light/8 h dark photoperiod. Following germination, each seedling was then transferred to a 750 mL plastic pot containing sterile soil. Approximately 30 d later, the rice plants were used for SBPH infestation. SBPH colonies were obtained from rice fields in Nanjing, China, and grown on rice seedlings in a climate chamber at 26 ± 1 °C with a 16 h light/8 h dark photoperiod.

### 4.2. Plants Infested by SBPH

Individually potted rice plants were placed in glass cylinders (2 cm diameter × 8 cm height) that surrounded the basal stem of each plant. Thirty fourth-instar SBPH nymphs were placed on each rice plant and fed for 12, 24, 48, or 72 h. The outer two leaf sheath tissues were collected simultaneously at different feeding times for subsequent transcriptomic and metabolomic analyses; the uninfested leaf sheath samples (0 h) were used as controls. Each experiment included three biological replicates.

### 4.3. SBPH Bioassays

The choice test was conducted as follows [44]: two rice plants were potted side-by-side in the same plastic container, with each plant placed in glass cylinders (2 cm diameter × 8 cm height), which surrounded the basal stem of each plant. One was infested with 30 fourth-instar SBPH nymphs, and the other was used as a control. After 48 h of infestation, SBPHs were removed, and one pair of rice plants was confined in a glass cylinder (4 cm diameter × 8 cm height), into which 15 fourth-instar nymphs or adult females were released in the middle of the two plants (Figure 1A,B). Following that, SBPHs settling on each plant were counted at 2, 4, 8, 12, 24, and 48 h. The experiment was repeated 12 times with 15 insects per replicate. In the no-choice test, the fecundity of female adult SBPHs was measured as follows [45]: pre-infested SBPH nymphs and control rice plants were prepared using the same method as above. Each plant was then placed in glass cylinders (2 cm diameter × 8 cm height), which surrounded the basal stem of each plant; five female adults (three days after emergence) were released into the plants (Figure 1C). After two days, the number of eggs laid by the five adult females on each plant was counted under an Olympus SZ51 microscope (Olympus, Tokyo, Japan). The experiment was repeated 15 times, with five insects per replicate.

### 4.4. Library Preparation and Illumina Sequencing

The cDNA libraries were sequenced on the Illumina sequencing platform by Metware Biotechnology Co., Ltd. (Wuhan, China), and the detailed procedures are as stated here [46]. Total RNA was extracted and purified using the TRIzol reagent (Invitrogen, Thermo Fisher Scientific, Waltham, MA, USA). A total of 1 µg of RNA per sample was used as the input material for RNA sample preparation. Sequencing libraries were generated using the NEB Next^®^ Ultra RNA Library Prep Kit for Illumina^®^ (New England Biolabs [NEB], Ipswich, MA, USA) following the manufacturer’s instructions, and index codes were added to attribute sequences to each sample. Briefly, mRNA was purified from total RNA using poly T oligo-attached magnetic beads. Fragmentation was performed using divalent cations at elevated temperatures in the Next First-Strand Synthesis Reaction Buffer (5×). First-strand cDNA was synthesized using random hexamer primers and M-MuLV reverse transcriptase. Second-strand cDNA synthesis was performed using DNA polymerase I and RNase H. Remaining overhangs were converted into blunt ends via exonuclease/polymerase activities. Following adenylation of the 3′ ends of the DNA fragments, Next Adaptor (NEB) with a hairpin loop structure was ligated to prepare for hybridization. Polymerase chain reaction (PCR) was performed using Phusion High-Fidelity DNA polymerase, Universal PCR primers, and Index Primer. Eventually, the PCR products were purified, and library quality was assessed using the Agilent Bioanalyzer 2100 system (Agilent Technologies, Santa Clara, CA, USA). The samples were designated as follows: 0 h, trans0 (trans0-1, trans0-2, and trans0-3); 12 h, trans12 (trans12-1, trans12-2, and trans12-3); 24 h, trans24 (trans24-1, trans24-2, and trans24-3); 48 h, trans48 (trans48-1, trans48-2, and trans48-3); 12-trans, 24-trans, and 48-trans were denoted as trans12 vs. trans0, trans24 vs. trans0, and trans48 vs. trans0, respectively.

### 4.5. Transcriptome Assembly and Functional Annotation

We used HISAT v2.1.0 software to construct the index and compare clean reads to the Japonica reference genome (ftp://ftp.ensemblgenomes.org/pub/plants/release-52/fasta/oryzasativa/dna; accessed on 8 October 2022). The mapped reads from each sample were assembled and predicted using StringTie v1.3.4d [47]. Subsequently, all transcriptomes from all samples were merged to construct a comprehensive transcriptome. To determine the gene alignment and fragments per kilobase of the exon model per million mapped reads (FPKM), featureCounts v1.6.2 /StringTie v1.3.4d was used. DESeq2 v1.22.1/edgeR v3.24.3 were used to analyze the differential expression between the two groups, and the *p*-value was corrected using the Benjamini and Hochberg method [48,49]. The corrected *p*-value ≤ 0.05, and the absolute value of Log2 FC (fold-change) ≥ 1.0 were used as the thresholds for significant differential expression [48]. Gene Ontology (GO) and Kyoto Encyclopedia of Genes and Genomes (KEGG) pathway enrichment analyses of the differentially expressed genes (DEGs) were performed using the GOseq R package and KOBAS 3.0 software, respectively. The enrichment analysis was performed based on the hypergeometric test. For KEGG, the hypergeometric distribution test was conducted with the unit of pathway; for GO, it was performed based on the GO terms [50,51].

### 4.6. Sample Preparation for Widely Targeted Metabolomics

Sample extraction and quantification were performed by Wuhan Metware Biotechnology Co. Ltd. (Wuhan, China). The freeze-dried samples were crushed using a mixer mill (MM 400; Retsch Gmbh, Haan, Germany), and 100 mg powder was weighed and extracted in 70% aqueous methanol. Following centrifugation, the extracts were absorbed and filtered before ultraperformance liquid chromatography–tandem mass spectrometry (UPLC-MS/MS) analysis [52]. The samples were designated as follows: 0 h, meta0 (meta0-1, meta0-2, and meta0-3); 24 h, meta24 (meta24-1, meta24-2, and meta24-3); 48 h, meta48 (meta48-1, meta48-2, and meta48-3); 72 h, meta72 (meta72-1, meta72-2, and meta72-3), 24-meta, 48-meta, and 72-meta were denoted as meta24 0 vs. meta0, meta48 vs. meta0, and meta72 vs. meta0, respectively.

### 4.7. UPLC Conditions

The sample extracts were analyzed using a UPLC-ESI-MS/MS system (UPLC, Shim-pack UFLC SHIMADZU CBM30A system, www.shimadzu.com.cn/; MS, Applied Biosystems 4500 Q TRAP, www.appliedbiosystems.com.cn; accessed on 15 October 2022). The analytical conditions were as follows: UPLC: column, Waters ACQUITY UPLC HSS T3 C18 (1.8 µm, 2.1 mm × 100 mm); the mobile phase consisted of solvent A (pure water with 0.04% acetic acid) and solvent B (acetonitrile with 0.04% acetic acid). Sample measurements were performed using a gradient program that employed starting conditions of 95% A and 5% B. Within 10 min, a linear gradient of 5% A and 95% B was programmed, and a composition of 5% A and 95% B was kept for 1 min. Subsequently, a composition of 95% A and 5.0% B was adjusted within 0.10 min and kept for 2.9 min. The column oven was set to 40 °C, and the injection volume was 4 μL. The effluent was alternatively connected to an ESI-triple quadrupole-linear ion trap (QTRAP)-MS [52,53].

### 4.8. Electrospray Ionization Triple Quadrupole Linear Ion Trap Tandem Mass Spectrometry (ESI-Q TRAP-MS/MS)

Lin-ear ion trap (LIT) and triple quadrupole (QQQ) scans were acquired on a triple quadrupole-linear ion trap mass spec trometer (Q TRAP), API 4500 Q TRAP UPLC/MS/MS System, equipped with an ESI Turbo Ion-Spray in terface, operating in positive and negative ion mode and controlled by Analyst 1.6.3 software (AB Sciex, Framingham, Massachusetts, USA). The ESI source operation parameters were as follows: ion source, turbo spray; source temperature 550 °C; ion spray voltage (IS) 5500 V (positive ion mode)/−4500 V (negative ion mode); ion source gas I (GSI), gas II (GSII), curtain gas (CUR) were set at 50, 60, and 30.0 psi, respectively; the collision gas (CAD) was high. Instrument tuning and mass calibration were performed with 10 and 100 μmol/L polypropylene glycol solutions in QQQ and LIT modes, respectively. QQQ scans were acquired as MRM experiments with collision gas (nitrogen) set to 5 psi. DP and CE for individual MRM transitions was done with further DP and CE optimization. A specific set of MRM transitions were monitored for each period according to the metabolites eluted within this period. Metabolite identification is based on the contain accurate mass of metabolites, MS2 fragments, MS2 fragments isotope distribution and retention time (RT). Using the company’s self-developed intelligent secondary spectrum matching method, the secondary spectrum and RT of the metabolites in the project samples are compared with MWDB (Metware database) and the public database of metabolite information (https://massbank.eu/MassBank/; https://pubchem.ncbi.nlm.nih.gov/, accessed on 11 October 2022), specifically, the MS tolerance and MS2 tolerance are set to 20 ppm, and RT offset does not exceed 0.2 min. Metabolites that do not have standard products will be compared with MS2 spectra in public databases or literature. Some of the metabolites without standard secondary spectra are inferred based on experience of mass spectrum analysis. The primary and secondary spectral data of mass spectrometry were qualitatively analyzed by software Analyst 1.6.3 and quantitatively analyzed by MRM [52,54,55].

### 4.9. Metabolite Profiling Analysis

Principal component analysis (PCA) was performed using the statistical function prcomp in R (www.r-project.org; accessed on 15 October 2022). The data were unit variance scaled prior to unsupervised PCA. The hierarchical cluster analysis (HCA) results for the samples and metabolites were presented as heatmaps, and the Pearson correlation coefficients (PCC) between samples were calculated using the cor function in R and presented as only heatmaps [52]. The variable importance in projection (VIP) ≥ 1.0 and the absolute value of Log_2_FC ≥ 1.0 were used as the thresholds for significantly regulated metabolites. The identified metabolites were annotated using the KEGG compound database (http://www.kegg.jp/kegg/compound; accessed on 16 October 2022), and the annotated metabolites were mapped to the KEGG pathway database (http://www.kegg.jp/kegg/pathway.html; accessed on 16 October 2022). Pathways enriched for significantly regulated metabolites were then fed into metabolite set enrichment analysis (MSEA) [56].

### 4.10. Quantitative PCR

Total RNA was isolated using the SV Total RNA Isolation System (Promega, Madison, WI, USA), according to the manufacturer’s instructions. After DNase treatment, 1 µg of total RNA was reverse-transcribed using the PrimeScript RT-PCR Kit (TaKaRa Bio Inc., Shiga, Japan). Quantitative PCR (qPCR) was performed using a TB GreenTM Premix Ex TaqTM Kit (TaKaRa) according to the manufacturer’s protocol and run in a LightCycler 480 II Real-Time System R (Roche Diagnostics, Basel, Switzerland). The internal normalization control was rice actin, and the qPCR primers are listed in Appendix A. The quantitative real-time PCR (RT-qPCR) was repeated three times, with three samples per replicate. Gene expression levels were calculated by normalizing target gene mRNA level to actin gene mRNA abundance using the 2^−ΔΔCt^ method.

### 4.11. Statistical Analysis

The following sample correspondence correlation analyses of transcriptomic and metabolomic data were performed: meta24-trans12 vs. meta0-trans0, meta48-trans24 vs. meta0-trans0, and meta72-trans48 vs. meta0-trans0. Based on correlation coefficient values of >0.8, a nine-quadrant plot of DEGs and differentially accumulated metabolites (DAMs) correlations was constructed. Duncan’s multiple range tests (or Student’s *t*-tests when only two treatments were compared) were used to investigate differences in the no-choice and qPCR tests, and χ^2^ tests were applied to the SBPH choice test. The statistical significance was determined using GraphPad Prism 8 (Version 8.0.2, GraphPad Software, La Jolla, CA, USA). * *p* < 0.05; ** *p* < 0.01. Statistica 6 (Statistica, SAS, Institute Inc., Cary, NC, USA) was used for data analysis.

## 5. Conclusions

This study revealed that SBPH nymph infestation makes rice more susceptible to adult SBPH infestation. We examined and discussed the functions of some potential target genes and metabolites associated with SBPH nymph resistance. Additionally, we made the assumption that a decrease in flavonoid accumulation in SBPH nymph-infested rice increases susceptibility to SBPH infestation. These findings add to our understanding of rice’s chemical defense against herbivores and suggest resistance genes and metabolites to improve the resistance of rice.

## Figures and Tables

**Figure 1 ijms-24-04764-f001:**
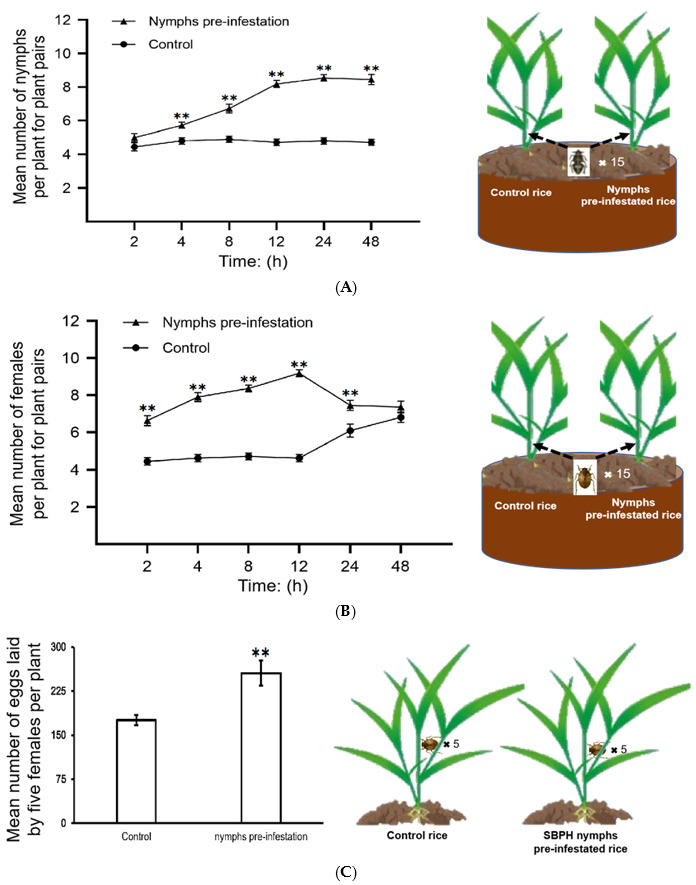
SBPH nymph infestation contributes to SBPH feeding and survival on rice. All of the choice and no choice assays for SBPH were performed after 48 h of continuous nymph pre-infestation; untreated SBPH was used as the control. A and B, SBPH host preference in choice tests. The mean number + SE (*n* = 15) of fourth-instar SBPH nymphs (**A**) and female adults (**B**) per plant for plant pairs (SBPH nymphs pre-infested rice vs. control). *n*, number of biological replicates. Fifteen fourth-instar nymphs or female adults were released per replicate, and the experiment was repeated 12 times. Statistical significance was calculated using GraphPad Prism 8 (Version 8.0.2, GraphPad Software, La Jolla, CA, USA). ** *p* < 0.01, Student’s *t*-tests. (**C**) Mean number of eggs laid by five female adults that fed on the control or SBPH nymph pre-infested rice plants for two days; the experiment was repeated 15 times. ** *p* < 0.01, Student’s *t*-tests. The model figures in the right half of the graph depict the SBPH choice and no choice assays.

**Figure 2 ijms-24-04764-f002:**
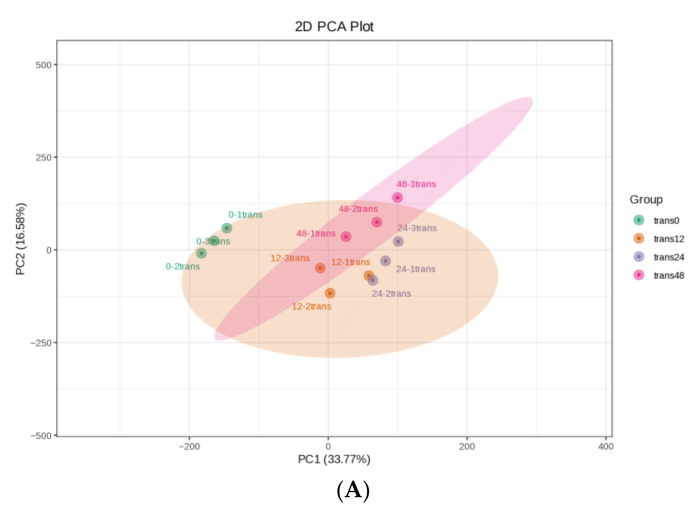
Overall transcriptomic changes in rice in response to SBPH nymph infestation. (**A**) Principal component analysis of each transcriptome sample; the X-axis and Y-axis represent the first and second principal components, respectively. (**B**) Hierarchical cluster analysis of differentially expressed genes (DEGs) between the three groups of SBPH nymphs-infested time-points (trans12, trans24, or trans48) and the uninfested (trans0) group, with each group containing three biological replicates. (**C**) Histogram statistics of DEGs in plants infested by SBPH nymphs (12 h time-point, 24 h time-point, or 48 h time-point) compared to the control group (0 h time-point). Total, all DEGs; own, down-regulated DEGs; p, up-regulated DEGs. (**D**) Venn of DEGs in three comparison groups.

**Figure 3 ijms-24-04764-f003:**
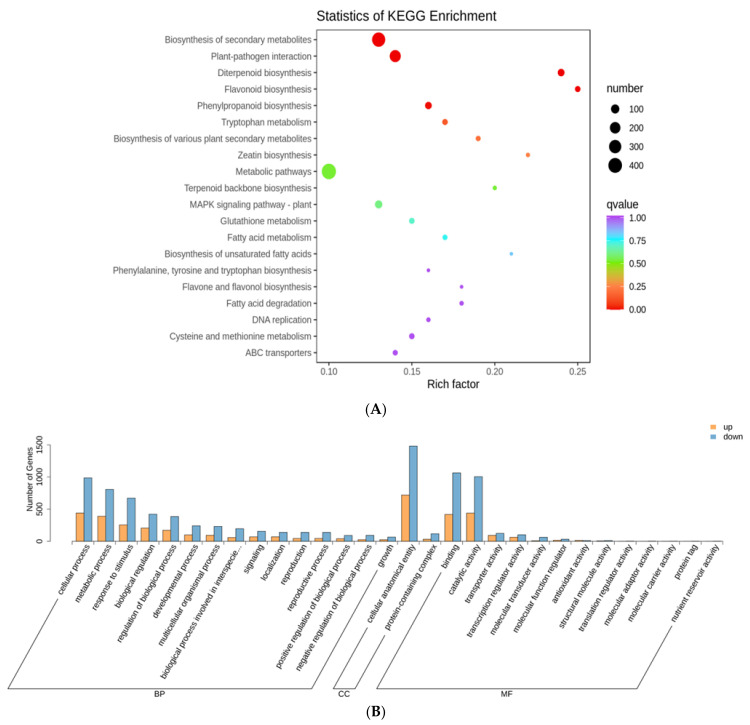
Kyoto Encyclopedia of Genes and Genomes (KEGG) pathways and Gene Ontology (GO) terms enriched for transcriptome DEGs in rice infested by SBPH nymph compared with untreated control. (**A**) A bubble plot with KEGG pathways enriched for DEGs in rice infested by SBPH nymphs at the 24 h time-point compared with the 0 h time-point. (**B**) Histogram with GO terms enriched for DEGs in rice infested by SBPH nymphs at the 24 h time-point compared with the 0 h time-point, *p* ≤ 0.05.

**Figure 4 ijms-24-04764-f004:**
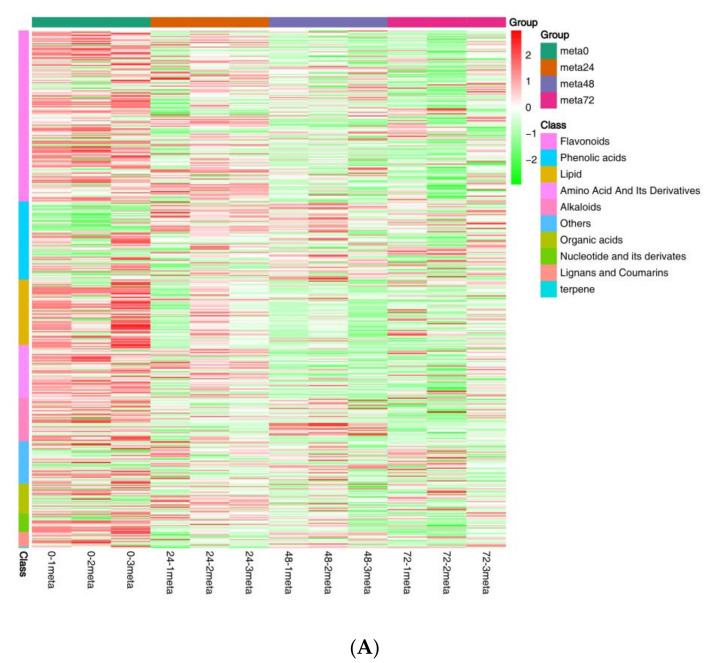
Overall analysis of the metabolomic changes in rice in response to SBPH nymph infestation. (**A**) Hierarchical cluster analysis of differentially accumulated metabolites (DAMs) between the three groups of SBPH nymphs-infested time-points (meta24, meta48, or meta72) and control (meta0) groups; each group contained three biological replicates. (**B**) Principal component analysis of each metabolite sample; the X-axis and Y-axis represent the first and second principal components, respectively. (**C**) A histogram with DAMs in plants infested by SBPH nymphs (24 h time-point, 48 h time-point, or 72 h time-point) compared to the control group (0 h time-point). Down, down-regulated DEGs; Up, up-regulated DEGs. (**D**) Venn of DAMs in each comparison group.

**Figure 5 ijms-24-04764-f005:**
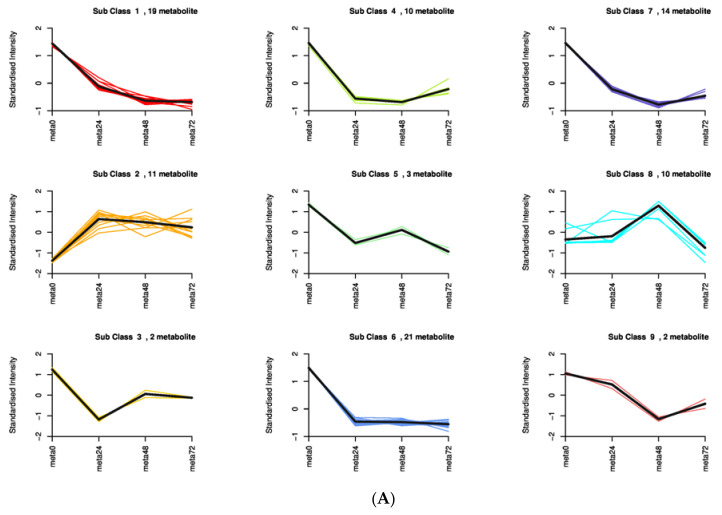
K-means plot and KEGG pathways enriched for metabolome DAMs in rice infested by SBPH nymphs compared with the untreated control. (**A**) The DAMs data from samples collected 0 h, 24 h, 48 h, and 72 h after treatment (meta0, meta24, meta48, and meta72) were used for the K-means plot. (**B**) KEGG pathways enriched for DAMs in rice infested by SBPH nymphs at the 48 h time-point compared with the 0 h time-point.

**Figure 6 ijms-24-04764-f006:**
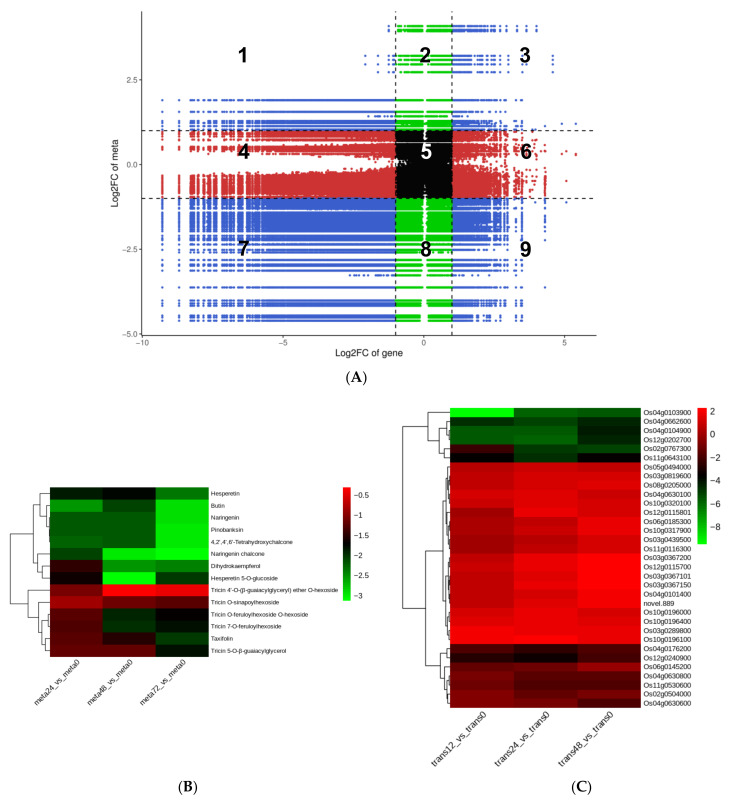
Correlation analysis revealing the potential regulatory network between DEGs and DAMs. (**A**) A nine-quadrant diagram showing the association of differentially expressed genes (DEGs) with differentially accumulated metabolites (DAMs) in the meta48-trans24 vs. meta0-trans0 comparison group. A Pearson correlation coefficient (PCC) of > 0.80 and *p* ≤ 0.05 were selected as the criteria and visualized using a nine-quadrant graph. The X-axis and Y-axis represent the log_2_FC of genes and metabolites, respectively. From left to right, from top to bottom, the nine-quadrant diagram was divided into 1–9 quadrants using a black dotted line. Quadrants 1–9 depict that the differential expression patterns of genes and metabolites are opposite; quadrant 5 shows that neither genes nor metabolites are differentially expressed; quadrants 2, 4, 6, and 8 indicate that metabolite expression is unchanged while genes are up or downregulated, or gene expression is unchanged while metabolites are up or downregulated; quadrants 3 and 7 show that the differential expression patterns of genes and metabolites are consistent. Heatmaps of DAMs (**B**) and DEGs (**C**) involved in the flavonoid biosynthesis pathway in the meta48-trans24 vs. meta0-trans0 comparison group. The bar at right represents the color code for Log_2_-transformed data on metabolite accumulation; red indicates higher expression, and blue indicates lower expression, *p* ≤ 0.05.

## Data Availability

The data that support the findings of this study are available from the corresponding author upon reasonable request.

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
