# Peer review of "Small Brown Planthopper Nymph Infestation Regulates Plant Defenses by Affecting Secondary Metabolite Biosynthesis in Rice"

_ijms, 2023, doi:10.3390/ijms24054764_

Round 1
Reviewer 1 Report
Comments:
The authors studied insect resistance in rice by focusing on SBPH (a type of insect) and attempting to identify the genes and metabolites that play a role in resistance to this insect. They used nymphs of SBPH to pre-infest rice plants and found that this increased the plants' susceptibility to subsequent herbivory by these insects. To identify the potential genes involved in this process, the authors analyzed the transcriptome and metabolome of the pre-infested samples and identified candidate genes and metabolites.
There are few comments to improve this study:
I understand the pre-treatment of rice plants with nymphs, I am curious why authors did not use adult SBPH for pre-infestation? Authors should explain this somewhere in introduction or in discussion.
Line 49. Authors can add more detail or rearrange introduction, for example, how secondary metabolites are involved in plant defense.
Line 78-86. Authors have performed choice and no choice assay for SBPH, It will be good for understanding of common readers if authors could add model figure showing the preference experiment. Overall, please improve the figure caption. Figure caption should be stand alone. Could you please improve the labelling in figure 1a and figure 1b. Its not clear from Y-axis what did you recorded?? Did you record number of insects attractive to pre-infested plants? ….
Line 93. Where is transcriptome data for time point 72h?? I can see metabolite data, but not for transcriptome…
Authors should explain how many DEGs were used for KEGG analysis.
Could you please adjust the font size labelling in figure 2? Some of fonts are smaller and some of them are greater. Font size should be consistent.
Line 140. I would suggest, please don’t use uncommon abbreviate. It will make problem for readers.
Line 177. Authors claim that most 177 DEGs were consistent with the DAMs patterns; some single DAMs were regulated by 178 multiple DEGs or a single DEG regulated multiple DAMs. Could you please explain which ones DEGs are consistent with DAMs? And which single DAMs were regulated by 178 multiple DEGs or a single DEG regulated multiple DAMs. Please mention names in the result section.
Line 180-82. Please also describe meaning of other quadrant (1-6 and 8).
Line 184. Please write complete sentence instead of “writing above mentioned”. Please mention time point and names of DEGs and DAMs KEGG 184 pathways?????
Line 186- 188. Please write the reference of this sentence. Where did you get this information? Figure ????
Line 189. Which results shows??? Please write complete information. For XXX, XXX and XXX revealed flavonoid compounds decrease whereas alkaloid …… Reff (Figure XX)
Line 189. Your rationale are not cleared here: we focused on the differences in 190 the expression of genes relevant ……
Line 247-253. Why did you choose genes for qPCR? Why did not choose well know marker genes for qPCR? For example, JA, SA, IAA, and biosynthesis genes as a marker?
I would suggest put these qPCR figures in supplementary data.
Line 280, 285, 304, 321. Remove word “ Figure X”
Line 315. Could you please make a table of data used in figure 4a and put in a supplementary table. I am really confused about the figure 4a. there is no labelling and legend. How to read it? Could you please provide more information in this figure? Please write quadrant numbers in the figure. Then describe them in figure caption. There are a lot of papers reporting this kind of figure, please consult …
Line 304-309. Please divide figure 3 into two figures. Figure 3abcd and Figure X ab.
Line 311. Please mention what kind of statistical analysis have been done???
Line 312. Please correct spelling “Wayne” into Venn.
Line 315-321. Could you please figure caption at the bottom of the figure 4?
Line 487. Could you please explain more about your RNA- Seq analysis methods? Did you both Stingtie and feature count or you used Hisat2 and then you had used feature count? We normally used Stringtie for assembly of transcriptome in de novo analysis.
Reviewer 2 Report
The manuscript titled “Small Brown Planthopper Nymph Infestation Regulates Plant Defenses by Affecting the Biosynthesis of Secondary Metabolites in Rice” is an interesting approach to decipher priming with plant and insect pest interaction. The results are well presented. Therefore, I think it can be accepted after correct some minor comments.

Reviewer 3 Report
The manuscript submitted by Li et al. describes the effect of feeding of a specific small brown planthopper developmental stage (nymph) on plant defense metabolism. Interestingly, nymph feeding increased subsequent grasshopper feeding on rice, though in many species the opposite is observed. The higher susceptibility correlated with a decrease in flavanoids.
The study is interesting and it is valuable that the authors tried to combine transcriptomic and metabolomic data. Nevertheless, there are still a number of points that need to be addressed.
It remains unclear how the authors annotated the metabolites found in this study. They speak of a targeted approach but the only reference provided is from a breast cancer study describing the general approach, but more details should be given regarding the confidence of identification based on the data-bases used. Furthermore, based on the short description provided and the reference it rather seems that the authors used an untargeted approach, not a targeted approach as mentioned in the abstract.
Furthermore, please provide a better description of the samples that you compare. It is fine to label groups with trans and meta in a figure and explain their meaning in the figure legends. However, the text gets much more readable if you for example clearly say - we compared the 24 h time-point with the controls at time 0 (instead of trans0, trans24) or we compared the metabolite data from samples collected 48 h after treatment with transcriptomic data generated 24 h after treatment (instead of meta48 vs. trans24).
The results of the study are correlative and therefore, conclusions remain partially speculative. Please mention in the discussion which further experiments are required to test your hypothesis.
It is good that you validated the transcriptome data by qpcr. Nevertheless, please move Figure 5 to Supplementals and refer to it in subchapter 2.2. Furthermore, it would be even better if the major findings could be confirmed in an independent set-up and not only be based on a single experiment.
Line 152-156 - I cannot follow this sentence. The alkaloids accounted for 50% of the total number of annotated metabolites. However, my understanding was, that at the 48 h time-point only 14 metabolites were significantly up-regulated but another 64 down-regulated. This should be clarified in the text.
The subheadings in the discussion are not very telling and not appropriate for a discussion. Please don't repeat the result but summarize what you discuss.
Titles of captions should be improved - instead of a generic sentence it would be good to summarize the findings shown in the figure in a single sentence for the figure caption and describe in more detail what is shown in the figure legend.
Minor comments:
Please write all species names in italics.
The manuscript contains too many abbreviations the reader has to remember throughout the text. This is not helpful. Please carefully go through the manuscript and reduce their number. Some abbreviations are only used once or twice, others such as PA are not explained, and some are used very often but it makes the text really hard to read.
Instead of listing 16 metabolites in the text in chapter 2.4 it would be better to focus on the most important ones and refer to a table.
Please move table 1 to Supplementals
Line 47 are (not is)
Line 74 speculated about the
Line 208 was gradually more pronounced
Line 351 delete PAs once
Line 367 PA content
Line 388 flavanoid content
Line 408 what is the difference between stress and metabolism proteins.
Round 2
Reviewer 1 Report
Dear Authors,
Thank you for following the suggestions. I dont have more comments. please have a look and improve minor mistakes in the MS. A few comments are here:
Line 16. Please add one word “crop” in the end of this sentence. which is the world’s major grain to “which is the world’s major grain crop”.
Line 21-23. please revise the sentence, it’s a too long sentence. “We observed that SBPH feeding induced significant changes in the levels of 92 metabolites, including 56 defense-related secondary metabolites (34 flavonoids, 17 alkaloids, and 5 phenolic acids), with more downregulated differentially accumulated metabolites than upregulated ones”.
Line 34. Change “ Rice production” into “The rice production”
Reviewer 3 Report
The authors addressed most of the points raised by the reviewers very well. There is only one aspect still missing - how did the authors annotate the metabolites. Two new references are provided but the authors should briefly summarize what they did.
There are a few minor errors:
Line 54 constitutively (not naturally)
Line 74 adult infestation
line 222 and (instead of nevertheless)
line 250 naringenin,2C2-oxoglutarate 3-dioxygenase
line 313 nymphs
lines 355 and 359 PAs (the abbreviation was introduced and used earlier in the text)
